# Your Classifier can Secretly Suffice Multi-Source Domain Adaptation

**Naveen Venkat**    **Jogendra Nath Kundu**    **Durgesh Kumar Singh**
**Ambareesh Revanur**    **R. Venkatesh Babu**
Video Analytics Lab, Indian Institute of Science, Bangalore
Corresponding author: `nav.naveenvenkat@gmail.com`

## Abstract

Multi-Source Domain Adaptation (MSDA) deals with the transfer of task knowledge from multiple labeled source domains to an unlabeled target domain, under a domain-shift. Existing methods aim to minimize this domain-shift using auxiliary distribution alignment objectives. In this work, we present a different perspective to MSDA wherein deep models are observed to implicitly align the domains under label supervision. Thus, we aim to utilize implicit alignment without additional training objectives to perform adaptation. To this end, we use pseudo-labeled target samples and enforce a classifier agreement on the pseudo-labels, a process called Self-supervised Implicit Alignment (SImpAl). We find that SImpAl readily works even under category-shift among the source domains. Further, we propose classifier agreement as a cue to determine the training convergence, resulting in a simple training algorithm. We provide a thorough evaluation of our approach on five benchmarks, along with detailed insights into each component of our approach.

## 1 Introduction

The task of supervised learning for classification is based on the assumption that the training data and the testing data are sampled from the same distributions. Thus, supervised learning methods achieve state-of-the-art results when evaluated on popular benchmarks such as ImageNet [46]. However, when such models are deployed in real-world, they yield sub-optimal results due to the inherent distribution-shift (domain-shift [55]) between the training data and the real-world environment (*a.k.a.* the target domain). While it is possible to obtain unlabeled samples from the target domain in most cases, the huge costs of data annotation prohibit the creation of a reliable labeled training dataset. To this end, Unsupervised Domain Adaptation (DA) methods have been proposed that aim to transfer knowledge from a labeled "source" dataset to an unlabeled "target" dataset under a domain-shift.

A popular strategy in Unsupervised DA is to learn the task-specific knowledge using supervision from the labeled source dataset, while learning a domain-invariant latent space where the features across the source and the target domains align. Such an alignment is enforced using statistical discrepancy minimization schemes [1, 12, 39, 43, 54] or via an adversarial objective [11, 30, 57, 61, 66], or by employing domain-specific transformations [6, 26, 44]. This alignment minimizes the domain-shift in the latent space, and improves the target generalization. However, the performance of Single-Source Domain Adaptation (SSDA) methods is usually determined by the choice of the source dataset [24].

Recently, Multi-Source Domain Adaptation (MSDA) [35, 67] has garnered interest wherein multiple labeled source domains are used to transfer the task knowledge to the unlabeled target domain. A common approach [15, 43, 61] is to learn a shared feature extractor, along with domain-specific classifier modules (Fig. 1a), which yield an ensemble prediction for the target samples. However, an additional challenge in MSDA is to tackle the domain-shift and category-shift [61] between each pair of source-domains (Fig. 1b). To this end, auxiliary losses are enforced encouraging the model to

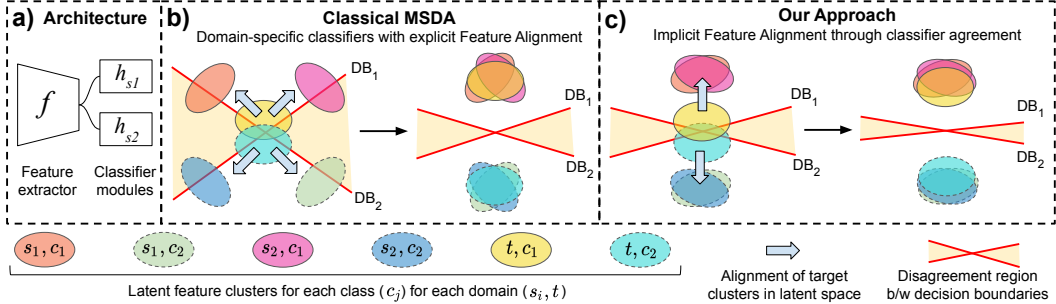

Figure 1: An illustration of the proposed concept for two-class ($c_1, c_2$) classification, with two labeled source domains ($s_1, s_2$) and one unlabeled target domain ($t$). Best viewed in color. **(a) Architecture.** Several works employ a shared feature extractor ($f$), and source-specific classifier modules. **(b) Classical MSDA.** Prior works employ source-specific classifiers that learn distinct (domain-specific) decision boundaries (denoted as $DB_i$). This results in a large discrepancy in the classifier predictions (region shaded in yellow). Thus, an auxiliary feature alignment loss is required for improving the classifier predictions. **(c) Our approach.** We enforce classifier agreement on source-domain samples leading to an *implicit alignment* of latent features. Further, imposing an agreement on the pseudo-labeled target samples improves the generalization to the target domain.

learn domain invariant but class-discriminative representations. Ultimately, an appropriate alignment of all the domains in the latent space [43] improves the generalization on the target domain (Fig. 1b).

In this work, we approach the MSDA problem from a different perspective. Since deep models are known to capture rich transferable representations [29, 38, 62], we ask, *is an auxiliary feature alignment loss really necessary*? The motivation stems from the observation that deep models exhibit a strong *inductive bias* to implicitly align the latent features under supervision. This is demonstrated in Fig. 2. Following the prior approaches [43, 61], we train domain-specific classifiers (Fig. 1b) and observe that the domains do not align in the latent space (Fig. 2a), which calls for an explicit feature alignment loss. However, when we enforce a classifier agreement on the class label for each input instance (Fig. 2b), we find that the domains tend to align, without requiring an explicit alignment loss.

This motivates us to further explore *implicit alignment* of latent features for MSDA. We aim to leverage the labeled data from multiple source domains, and the multi-classifier setup (Fig. 1a) employed in MSDA to perform alignment, without incorporating auxiliary components such as a domain discriminator [61, 66]. In contrast to learning domain-specific classifier modules, we enforce an agreement among the classifiers (Fig. 1c) to align the domains in the latent space.

Since the target domain is unlabeled, we resort to the class labels predicted by the model being trained (*a.k.a.* pseudo-labels [25]). The adaptation step encourages the classifiers to agree upon these pseudo-labels which enables alignment of the target features with the source features that have classifier agreement owing to label supervision. Accordingly, we name the approach as **S**elf-supervised **Imp**licit **Al**ignment, abbreviated as SImpAl (pronounced "simple"). We observe that even under category-shift, implicit alignment can be leveraged to align the shared categories, without requiring additional components (*e.g.* fine-grained alignment [5, 22, 42], adversarial discriminator [61]) or cumbersome training strategies (*e.g.* to handle arbitrary category-shifts [23, 61, 63]). We also find that classifier agreement can be leveraged as a cue to determine adaptation convergence.

To summarize, we demonstrate successful MSDA by leveraging implicit alignment exhibited by deep classifiers, corroborating the potential for designing simple and effective adaptation algorithms. We conduct extensive evaluation of our approach over five benchmark datasets, with two popular CNN backbone models (ResNet-50, ResNet-101 [16]) and derive insights from the empirical analysis.

## 2 Related Work

Here, we briefly review the related works and refer the reader to [67] for an extensive survey.

**a) Single-Source Domain Adaptation (SSDA).** Motivated by the seminal work by Ben-David *et al.* [2, 3], a large number of SSDA methods [6, 10, 11, 12, 28, 29, 32] have been proposed, that aim to learn domain-agnostic but class-discriminative representations. Inspired by the GAN framework [13], a popular strategy is to employ adversarial learning [18, 20, 51, 52, 56, 57, 58] that aims to confuse a

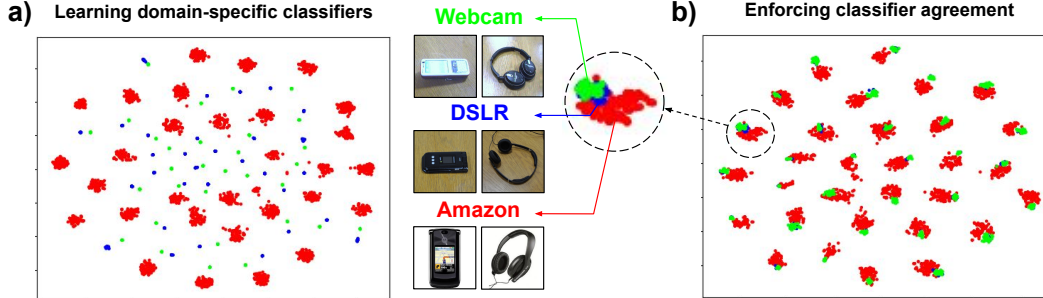

Figure 2: t-SNE plot of the features at the pre-classifier space, showing the feature distribution after learning a fully supervised model. Best viewed in color. **(a) Learning domain-specific classifiers.** We train a model (with ResNet-50 [16] backbone) with full label supervision from all the three domains on Office-31 [47], while keeping the classifier heads unique to each domain. Although we find that class discrimination is achieved, each domain forms separate sub-clusters, and does not align in the latent space. **(b) Enforcing classifier agreement.** Instead of learning domain-specific classifiers, we enforce the classifiers to agree upon the labels for all the samples. We observe that all the domains tend to align, without enforcing an explicit alignment objective, even under a domain-shift. We aim to leverage this inductive bias of deep models, to perform adaptation.

domain-discriminator, thereby aligning the latent features of the domains. Saito *et al.* [50] formulate an adversarial objective employing classifier discrepancy. In contrast, we aim to study a simpler approach which circumvents the training difficulties encountered in adversarial learning paradigms. Recently, consistency based regularizers [8, 21, 36, 20] were proposed for domain adaptation. In our work, classifier agreement can be interpreted as a form of consistency at the output space which acts both as an implicit regularizer and as a means to perform latent space alignment for adaptation.

**b) Multi-Source Domain Adaptation (MSDA).** Several methods [15, 43, 61, 68] learn domain-specific classifier modules and obtain a weighted ensemble prediction for the target samples, motivated by the distribution weighted combining rule [17, 34, 35]. Zhe *et al.* [68] employ an alignment loss between each source-target pair in domain-specific feature spaces. In addition, Peng *et al.* [43] align each pair of source domains using kernel based moment matching and also propose a variant based on adversarial learning [50]. Xu *et al.* employ multiple domain discriminators to achieve latent space alignment. In this work, we aim to explore a simple adaptation scheme that leverages implicit alignment in deep models. As a result, our approach is applicable even under category-shift among the source domains, while most prior methods [15, 43, 68] consider only a shared category set.

**c) Self-training methods.** Pseudo-labeling [25] is a popular semi-supervised learning approach where "pseudo" class labels are assigned to unlabeled samples, typically using classifier confidence [7, 49, 61, 69, 70] or nearest neighbor assignment [22, 40, 48, 65], while the model is retrained using such samples. Confidence thresholding [27, 49, 61] is commonly applied to minimize the noise in pseudo-labels. This introduces a sensitive threshold hyperparameter, requiring labeled target samples or domain expertise for precise tuning. Works such as Zou *et al.* [69, 70], Li *et al.* [27] and Chen *et al.* [8] propose various regularizers to improve pseudo-label predictions. Xu *et al.* [61] incorporate an adversarial alignment loss to mitigate the performance degradation arising from noisy pseudo-labels. In contrast, we aim to exploit classifier agreement to perform adaptation and improve the reliability of pseudo-labels without incorporating additional hyperparameters.

## 3 Self-supervised Implicit Alignment (SImpAl)

**Notations.** Let $\mathcal{X}$ and $\mathcal{Y}$ denote the input and the output spaces. We consider $n_d$ labeled source domain datasets $\{\mathcal{D}_{s_i}\}_{i=1}^{n_d}$, where $\mathcal{D}_{s_i} = \{(\mathbf{x}_{s_i}^{c_j}, y_{s_i}^{c_j}) \in \mathcal{X} \times \mathcal{Y}\}$ and a single unlabeled target domain dataset $\mathcal{D}_t = \{\mathbf{x}_t \in \mathcal{X}\}$. Each source domain has a label-set $\mathcal{C}_{s_i}$, and the target label-set is defined as $\mathcal{C} = \cup_{i=1}^{n_d} \mathcal{C}_{s_i}$ with $n_c$ classes. We learn a deep neural network model having a CNN based feature extractor $f : \mathbb{R}^{224 \times 224 \times 3} \to \mathbb{R}^{256}$, and $n_d$ classifier modules $h : \mathbb{R}^{256} \to \mathbb{R}^{n_d \times n_c}$. For convenience, we denote the output of the network as a matrix $\mathbf{M} = h \circ f(\mathbf{x})$, where $\circ$ represents function composition. $\mathbf{M}$ is obtained by stacking the logits produced by each classifier (see Fig. 3).

**Overview.** As is conventional in the MSDA methods [15, 61], the multi-classifier setup is treated as an ensemble of diverse classifiers, and the class probabilities are obtained through a convex

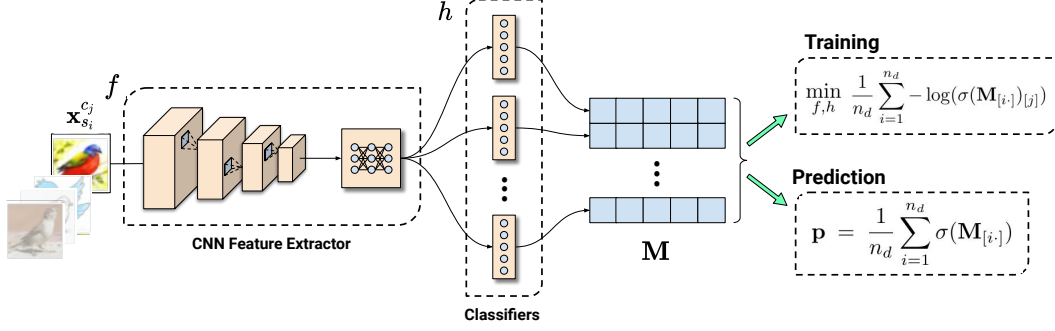

Figure 3: Architecture of the proposed approach. The network contains a common feature extractor $f$ having a CNN backbone followed by fully-connected layers. The multi-classifier module $h$ contains $n_d$ classifiers.

combination of each classifier's prediction. The model is first trained with the categorical cross-entropy loss imposed on the combined data from all source domains. After a "warm-start", we introduce pseudo-labeled target samples into the training process. The adaptation is performed by enforcing the classifiers to agree on these pseudo-labels. We now describe the approach in detail.

## 3.1 Warm-start with source domains

To adapt the network to the target domain, we use pseudo-labeled target samples. Thus, we first aim to achieve a reliability in pseudo-labels by training the model on all source domains. We call this as the warm-start process, which is performed as follows.

**a) Learning with source domains.** For each source-domain instance $\mathbf{x}_{s_i}^{c_j}$, we obtain the output matrix $\mathbf{M} = h \circ f(\mathbf{x}_{s_i}^{c_j})$ (see Fig. 3) and define the class probability vector $\mathbf{p}$ as a convex combination of the probabilities assigned by each classifier,

$$\mathbf{p} = \frac{1}{n_d} \sum_{i=1}^{n_d} \sigma(\mathbf{M}_{[i\cdot]}) \tag{1}$$

where, $\mathbf{M}_{[i\cdot]}$ represents the $i^{\text{th}}$ row vector of the matrix $\mathbf{M}$ (*i.e.* the logits of the $i^{\text{th}}$ classifier), and $\sigma(\mathbf{v})_{[j]} = \exp(\mathbf{v}_{[j]})/\sum_{j'=1}^{n_c} \exp(\mathbf{v}_{[j']})$ is the softmax function. Treating $\mathbf{p}$ as the class probability vector, we minimize the categorical cross entropy loss ($l_{ce}$) using the labeled source samples,

$$l_{ce}(\mathbf{p}, y_{s_i}^{c_j}) = -\log(\mathbf{p}_{[j]}) = -\log\left(\frac{1}{n_d} \sum_{i=1}^{n_d} \sigma(\mathbf{M}_{[i\cdot]})_{[j]}\right) \leq \frac{1}{n_d} \sum_{i=1}^{n_d} -\log(\sigma(\mathbf{M}_{[i\cdot]})_{[j]}) \tag{2}$$

The last term in Eq. 2 represents an upper bound for the categorical cross-entropy loss of the ensemble, and is obtained by applying the Jensen's inequality for convex functions. We consider the formulation in Eq. 2 to drive the classifiers to agree upon the label $y_{s_i}^{c_j}$ for $\mathbf{x}_{s_i}^{c_j}$. Thus, the training objective is,

$$\min_{f,h} \; \mathop{\mathbb{E}}_{\mathcal{D} \in \{\mathcal{D}_{s_{i'}}\}_{i'=1}^{n_d}} \; \mathop{\mathbb{E}}_{(\mathbf{x}_{s_i}^{c_j}, y_{s_i}^{c_j}) \in \mathcal{D}} \; \frac{1}{n_d} \sum_{i=1}^{n_d} -\log(\sigma(\mathbf{M}_{[i\cdot]})_{[j]}) \tag{3}$$

The objective in Eq. 3 is minimized by mini-batch stochastic optimization. Each mini-batch contains an equal number of samples from each source domain. In practice, each classifier is given a distinct random initialization, and is trained with the same set of training samples at each mini-batch. Intuitively, this process gradually enables a higher degree of similarity among the classifiers (Fig. 1c) through an agreement in the predicted class labels for source samples. Note that, both the feature extractor $f$ and the multi-classifier module $h$ are shared across all source domains. This step provides a warm-start to introduce pseudo-labeled target samples into the training.

**b) Determining the convergence of warm-start.** The next question we address is, *how to tell if a model is trained sufficiently for the target domain?* Intuitively, we would like to train the model until there is a saturation in the target (pseudo-label) accuracy. However, with unlabeled target samples

measuring the pseudo-label accuracy is out of bounds. Thus, we propose the classifier agreement as a criterion to determine the convergence. The classifier agreement for an instance $\mathbf{x}$ is defined as,

$$a(\mathbf{x}, f, h) = \prod_{i \neq i'} I( \underset{j \in \mathcal{C}}{\arg \max} \, \mathbf{M}_{[ij]} = \underset{j \in \mathcal{C}}{\arg \max} \, \mathbf{M}_{[i'j]} ) \qquad (4)$$

where $\mathbf{M} = h \circ f(\mathbf{x})$, and $I(\cdot)$ is the indicator function that returns 1 when the condition is true, else returns 0. Intuitively, when each classifier predicts the same class for a given sample $\mathbf{x}$, we say that the classifiers "agree". Thus, $a(\mathbf{x}, f, h) = 1$ when classifiers agree, and $a(\mathbf{x}, f, h) = 0$ otherwise.

As we shall show in Sec. 4.2, the target pseudo-label accuracy is higher whenever the classifiers agree [37, 64]. Thus, classifier agreement is used to filter out target samples having a higher degree of noise in pseudo-labels. Further, we estimate the fraction of target samples for which there is an agreement in the class predictions among the classifiers. Thus, we define the target agreement rate as,

$$A(\mathcal{D}_t, f, h) = \frac{1}{|\mathcal{D}_t|} \sum_{\mathbf{x}_t \in \mathcal{D}_t} a(\mathbf{x}_t, f, h) \qquad (5)$$

We hypothesize that the performance on target samples attains a saturation when the agreement rate converges. Thus, we determine the warm-start interval based on the convergence of $A(\cdot)$.

## 3.2 Introducing target data

After the warm-start, we introduce target samples into the training process. The pseudo-labels are obtained from the classifier predictions as in Eq. 1, *i.e.* $y_t^{c_j} = \arg \max_{j'} \frac{1}{n_d} \sum_{i=1}^{n_d} \sigma(\mathbf{M}_{[ij']})$.

We consider the following strategy for pseudo-labeling. To begin with, we select only those target samples for which there is a classifier agreement, since the labels are seen to be more accurate for such samples (verified in Sec. 4.2). Thus, we obtain a subset $\mathcal{D}_t' = \{(\mathbf{x}_t, y_t^{c_j}) \mid \mathbf{x}_t \in \mathcal{D}_t, a(\mathbf{x}_t, f, h) = 1\}$. Secondly, inspired by curriculum learning [4, 70] we form an easy-to-hard sampling strategy for $\mathcal{D}_t'$. For this purpose, we obtain the average classifier margin as a weight for each target instance,

$$w(\mathbf{x}_t, f, h) = \frac{1}{n_d} \sum_{i=1}^{n_d} (\mathbf{M}_{[ij]} - \mathbf{M}_{[ij']}) \qquad (6)$$

where $j$ an $j'$ correspond to the indices of the highest and the second highest logit. Intuitively, $w$ measures a form of confidence in prediction. Target samples that are farther from the decision boundaries receive a higher $w$ (see Fig. 7c for the geometrical interpretation). We show in Sec. 4.2 that, in general, samples with a higher $w$ are more likely to possess correct pseudo-labels. Thus, target samples are sorted based on $w$, and are fed to the training pipeline in the decreasing order of $w$. Finally, the pseudo-labels are updated every $n_e$ epochs on $\mathcal{D}_t'$. With this strategy, we formalize the training objective for adaptation using the target samples as,

$$\min_{f,h} \underset{(\mathbf{x}_t, y_t^{c_j}) \in \mathcal{D}_t'}{\mathbb{E}} \frac{1}{n_d} \sum_{i=1}^{n_d} -\log(\sigma(\mathbf{M}_{[i\cdot]})_{[j]}) \qquad (7)$$

After introducing the target samples from $\mathcal{D}_t'$, we train on both source and target samples, in alternate mini-batches, *i.e.* we minimize the objectives in Eq. 3 and Eq. 7 in alternate mini-batches. Finally, the network is trained until the target agreement rate $A$ shows convergence. This enables a simple and effective adaptation pipeline using implicit alignment. The algorithm is given in Algo. 1.

## 4  Experiments

We present the results of our approach on five standard benchmark datasets - *Office-Caltech*, *Image-CLEF*, *Office-31*, *Office-Home* and the most challenging large-scale benchmark, *DomainNet*.

**a) Prior Arts.** We compare against Deep Domain Confusion (DDC) [58], Deep Adaptation Network (DAN) [29], Deep CORAL (D-CORAL) [54], Reverse Gradient (RevGrad) [10], Residual Transfer Network (RTN) [32], Joint Adaptation Network (JAN) [31], Maximum Classifier Discrepancy (MCD) [50], Manifold Embedded Distribution Alignment (MEDA) [60], Adversarial Discriminative Domain Adaptation (ADDA) [57], Deep Cocktail Network (DCTN) [61], Moment Matching

---

**Algorithm 1** SImpAl - Self-supervised Implicit Alignment

---

1: **require:** Source datasets $\{\mathcal{D}_{s_i}\}_{i=1}^{n_d}$, Target dataset $\mathcal{D}_t$, Model $\{f, h\}$

2: **while** $A(\mathcal{D}_t, f, h)$ has not converged **do**                 ▷ Warm-start with source domains
3:     Load a mini-batch of samples $(\mathbf{x}_{s_i}^{c_j}, y_{s_i}^{c_j})$ from each source dataset $\mathcal{D}_{s_i}$
4:     Update $\{f, h\}$ using the objective in Eq. 3

5: Obtain pseudo-labeled target subset $\mathcal{D}_t' = \{(\mathbf{x}_t, y_t^{c_j}) \mid \mathbf{x}_t \in \mathcal{D}_t,\ a(\mathbf{x}_t, f, h) = 1\}$
6: Prepare $\mathcal{D}_t'$ by sorting the samples in descending order of $w(\mathbf{x}_t, f, h)$ (as in Eq. 6)

7: **while** $A(\mathcal{D}_t, f, h)$ has not converged **do**                 ▷ Introducing target samples

8:     Load a mini-batch of samples $(\mathbf{x}_{s_i}^{c_j}, y_{s_i}^{c_j})$ from each source dataset $\mathcal{D}_{s_i}$
9:     Update $\{f, h\}$ using the objective in Eq. 3

10:     Load a mini-batch of samples $(\mathbf{x}_t, y_t^{c_j})$ from pseudo-labeled target subset $\mathcal{D}_t'$
11:     Update $\{f, h\}$ using the objective in Eq. 7

12:     **if** $n_e$ epochs on $\mathcal{D}_t'$ are completed **then**        ▷ Periodically update pseudo-labels
13:         Perform steps 5-6 to recompute $\mathcal{D}_t'$

---

(M$^3$SDA) [43] and Multiple Feature Space Adaptation Network (MFSAN) [68]. Specifically, DDC, RevGrad, ADDA, MCD, DCTN use an adversarial alignment objective to perform adaptation, RTN learns a residual function to bridge the distribution discrepancy, and DAN, MFSAN, D-CORAL, JAN, MEDA and M$^3$SDA employ a kernel based moment matching scheme to align the domains.

**b) Evaluation.** For *ImageCLEF* and *Office*-based datasets, we follow the evaluation protocol in MFSAN [68], while for DomainNet, we follow the protocol used in M$^3$SDA [43]. Three types of baselines are considered - 1) Single Best (SB) refers to the best single-source transfer results for the target domain, 2) Source Combine (SC) refers to the scenario where all sources are combined into a single source domain to perform SSDA, 3) Multi-Source (MS) refers to the MSDA methods. We report the multi-run statistics (mean and standard deviation) obtained over three different runs.

**c) Implementation Details.** We implement our approach in PyTorch [41]. We use the Adam [19] optimizer, with learning rate $10^{-5}$ and weight decay $5 \times 10^{-4}$ for stochastic optimization. The losses in Eq. 3 and Eq. 7 are alternatively optimized and the target agreement rate (Eq. 5) is periodically monitored for convergence. We set $n_e = 15$ epochs as the update rate for the target pseudo-labels (line 12 in Algo. 1). The total number of training iterations are decided based on the convergence of the target agreement rate $A$ for each dataset. Following prior MSDA approaches [68, 43], we use ResNet-50 (SImpAl$_{50}$) and ResNet-101 (SImpAl$_{101}$) [16] as the CNN backbone. See code implementation[1] for architecture, hyperparameter values, and instructions to reproduce the results.

## 4.1 Results

We present the results in Table 1. The results for the prior baselines are reported from [43] and [68]. Due to the limits of space, we present the full comparison table for *DomainNet* in the Supplementary.

*Office-31* [47] dataset has 4652 images across Amazon (**A**), DSLR (**D**) and Webcam (**W**) domains having 31 object classes found in an office environment. *ImageCLEF*[2] dataset has been created by selecting 12 shared classes among *ImageNet* (**I**) [46], *Caltech-256* (**C**) [14], *Pascal-VOC 2012* (**P**) [9], with 600 images per domain. *Office-Caltech* [12] dataset consists of 2533 images across 10 classes shared between *Caltech-256* (**C**) and the three domains of *Office-31* (**A**, **D**, **W**). *Office-Home* [59] is a more challenging medium-scale dataset containing about 15588 images in 4 domains: Art (**Ar**), Clipart (**Cl**), Product (**Pr**) and Real-World (**Rw**), sharing 65 categories of objects found in the office and home environments. *DomainNet* [43] dataset is the largest and the most challenging benchmark, containing 6 diverse domains, with 345 classes, and around 0.6 million images.

Table 1: **Results on five standard benchmark datasets.** 'SB' stands for Single Best, 'SC' stands for Source Combined, and 'MS' denotes MSDA methods. The results for prior baselines are reported from [43] and [68]. See Supplementary for the full comparison table on DomainNet.

A. *Office-31*

| | Method | →D | →W | →A | Avg |
|---|---|---|---|---|---|
| SB | Source Only | 99.3 | 96.7 | 62.5 | 86.2 |
| SB | DDC | 98.2 | 95.0 | 67.4 | 86.9 |
| SB | DAN | 99.5 | 96.8 | 66.7 | 87.7 |
| SB | D-CORAL | 99.7 | 98.0 | 65.3 | 87.7 |
| SB | RevGrad | 99.1 | 96.9 | 68.2 | 88.1 |
| SB | RTN | 99.4 | 96.8 | 66.2 | 87.5 |
| SC | DAN | 99.6 | 97.8 | 67.6 | 88.3 |
| SC | D-CORAL | 99.3 | 98.0 | 67.1 | 88.1 |
| SC | RevGrad | 99.7 | 98.1 | 67.6 | 88.5 |
| MS | DCTN | 99.3 | 98.2 | 64.2 | 87.2 |
| MS | MFSAN | 99.5 | 98.5 | 72.7 | 90.2 |
| MS | $\text{SImpAl}_{50}$ | $99.2^{\pm0.2}$ | $97.4^{\pm0.1}$ | $70.6^{\pm0.6}$ | $89.0^{\pm0.3}$ |
| MS | $\text{SImpAl}_{101}$ | $99.4^{\pm0.2}$ | $97.9^{\pm0.2}$ | $71.2^{\pm0.4}$ | $89.5^{\pm0.3}$ |

B. *ImageCLEF*

| | Method | →P | →C | →I | Avg |
|---|---|---|---|---|---|
| SB | Source Only | 74.8 | 91.5 | 83.9 | 83.4 |
| SB | DDC | 74.6 | 91.1 | 85.7 | 83.8 |
| SB | DAN | 75.0 | 93.3 | 86.2 | 84.8 |
| SB | D-CORAL | 76.9 | 93.6 | 88.5 | 86.3 |
| SB | RevGrad | 75.0 | 96.2 | 87.0 | 86.1 |
| SB | RTN | 75.6 | 95.3 | 86.9 | 85.9 |
| SC | DAN | 77.6 | 93.3 | 92.2 | 87.7 |
| SC | D-CORAL | 77.1 | 93.6 | 91.7 | 87.5 |
| SC | RevGrad | 77.9 | 93.7 | 91.8 | 87.8 |
| MS | DCTN | 75.0 | 95.7 | 90.3 | 87.0 |
| MS | MFSAN | 79.1 | 95.4 | 93.6 | 89.4 |
| MS | $\text{SImpAl}_{50}$ | $77.5^{\pm0.3}$ | $93.3^{\pm0.3}$ | $91.0^{\pm0.4}$ | $87.3^{\pm0.3}$ |
| MS | $\text{SImpAl}_{101}$ | $78.0^{\pm0.5}$ | $95.2^{\pm0.5}$ | $91.7^{\pm0.4}$ | $88.3^{\pm0.5}$ |

C. *Office-Caltech*

| | Method | →W | →D | →C | →A | Avg |
|---|---|---|---|---|---|---|
| SC | Source Only | 99.0 | 98.3 | 87.8 | 86.1 | 92.8 |
| SC | DAN | 99.3 | 98.2 | 89.7 | 94.8 | 95.5 |
| MS | Source Only | 99.1 | 98.2 | 85.4 | 88.7 | 92.9 |
| MS | DAN | 99.5 | 99.1 | 89.2 | 91.6 | 94.8 |
| MS | DCTN | 99.4 | 99.0 | 90.2 | 92.7 | 95.3 |
| MS | JAN | 99.4 | 99.4 | 91.2 | 91.8 | 95.5 |
| MS | MEDA | 99.3 | 99.2 | 91.4 | 92.9 | 95.7 |
| MS | MCD | 99.5 | 99.1 | 91.5 | 92.1 | 95.6 |
| MS | M³SDA | 99.4 | 99.2 | 91.5 | 94.1 | 96.1 |
| MS | $\text{SImpAl}_{50}$ | $99.3^{\pm0.1}$ | $99.8^{\pm0.1}$ | $92.2^{\pm0.1}$ | $95.3^{\pm0.2}$ | $96.7^{\pm0.1}$ |
| MS | $\text{SImpAl}_{101}$ | $100^{\pm0.0}$ | $100^{\pm0.0}$ | $94.6^{\pm0.2}$ | $95.6^{\pm0.3}$ | $97.5^{\pm0.1}$ |

D. *Office-Home*

| | Method | →Ar | →Cl | →Pr | →Rw | Avg |
|---|---|---|---|---|---|---|
| SB | Source Only | 65.3 | 49.6 | 79.7 | 75.4 | 67.5 |
| SB | DDC | 64.1 | 50.8 | 78.2 | 75.0 | 67.0 |
| SB | DAN | 68.2 | 56.5 | 80.3 | 75.9 | 70.2 |
| SB | D-CORAL | 67.0 | 53.6 | 80.3 | 76.3 | 69.3 |
| SB | RevGrad | 67.9 | 55.9 | 80.4 | 75.8 | 70.0 |
| SC | DAN | 68.5 | 59.4 | 79.0 | 82.5 | 72.4 |
| SC | D-CORAL | 68.1 | 58.6 | 79.5 | 82.7 | 72.2 |
| SC | RevGrad | 68.4 | 59.1 | 79.5 | 82.7 | 72.4 |
| MS | MFSAN | 72.1 | 62.0 | 80.3 | 81.8 | 74.1 |
| MS | $\text{SImpAl}_{50}$ | $70.8^{\pm0.2}$ | $56.3^{\pm0.2}$ | $80.2^{\pm0.3}$ | $81.5^{\pm0.3}$ | $72.2^{\pm0.6}$ |
| MS | $\text{SImpAl}_{101}$ | $73.4^{\pm0.4}$ | $62.4^{\pm0.1}$ | $81.0^{\pm0.2}$ | $82.7^{\pm0.2}$ | $74.8^{\pm0.2}$ |

E. *DomainNet*

| | Method | →Clp | →Inf | →Pnt | →Qdr | →Rel | →Skt | Avg |
|---|---|---|---|---|---|---|---|---|
| MS | M³SDA | $57.2^{\pm0.9}$ | $24.2^{\pm1.2}$ | $51.6^{\pm0.4}$ | $5.2^{\pm0.4}$ | $61.6^{\pm0.9}$ | $49.6^{\pm0.5}$ | $41.5^{\pm0.7}$ |
| MS | $\text{SImpAl}_{101}$ | $66.4^{\pm0.8}$ | $26.5^{\pm0.5}$ | $56.6^{\pm0.7}$ | $18.9^{\pm0.8}$ | $68.0^{\pm0.5}$ | $55.5^{\pm0.3}$ | $48.6^{\pm0.6}$ |

## 4.2 Analysis

**a) Implicit alignment of features.** In Fig. 4a, we plot the t-SNE [33] embeddings of the features at the pre-classifier space (output of $f$) for SImpAl. Further, we calculate the Proxy-$\mathcal{A}$ distance [2] defined as $\text{dist}_{\mathcal{A}} = 2(1 - 2\epsilon)$ where $\epsilon$ is the generalization error of a domain discriminator. In Fig. 4b, we report the $\text{dist}_{\mathcal{A}}$ value across each source-target pair for 3 different models - 1) warm-start model, trained on the source domains, 2) the model after adaptation using SImpAl, 3) an oracle model employing SImpAl, where the target pseudo-labels are replaced by the ground-truth labels. This shows that adaptation using SImpAl effectively reduces the distribution-shift in the latent space. Further, we also demonstrate implicit alignment under large domain-shifts (such as Quickdraw and Real-world domains on DomainNet), which enables applications such as cross-domain image retrieval on an unlabeled target domain. See Suppl. for further analysis on implicit alignment.

**b) Extension to category-shift.** To present a more practical scenario for MSDA, [61] introduced two category-shift settings - overlap and disjoint, where the source domains contain overlapping label sets (*i.e.* $\mathcal{C}_{s_i} \cap \mathcal{C}_{s_{i'}} \neq \phi$, but $\mathcal{C}_{s_i} \cap \mathcal{C}_{s_{i'}} \neq \mathcal{C}_{s_i} \cup \mathcal{C}_{s_{i'}}$) and disjoint label-sets ($\mathcal{C}_{s_i} \cap \mathcal{C}_{s_{i'}} = \phi$) respectively. In such scenarios, it is vital to prevent mis-alignment of different classes across the source domains to avoid negative transfer [42]. Furthermore, since prior MSDA approaches learn domain-specific classifiers, they require separate mechanisms to obtain class probabilities for the domain-specific and the shared classes separately [61]. However, our approach remains unmodified under the presence of category-shift; as such, each classifier learns all the target classes, and the computation of the class probabilities (Eq. 1) remains unchanged. Fig. 4c shows that category-shift is a challenging scenario where all methods show performance degradation, however SImpAl is found to exhibit a relatively lower degradation in the target performance. This is supported by the observation that even under category-shift, only the shared classes align as shown in Fig. 5. See Suppl. for further analysis.

**c) Target Agreement Rate.** Fig. 6a shows the trend in the target agreement rate ($A(\mathcal{D}_t, f, h)$) and target performance as training proceeds. We make two observations. Firstly, we find that $A$ increases during training, indicating that the target samples migrate into the classifier agreement region in the latent space ($\{f(\mathbf{x}_t) \mid a(\mathbf{x}_t, f, h) = 1\}$). This migration is necessary for a successful adaptation since the source domains inherently fall in the classifier agreement region (due to the nature of the source training for warm-start). Secondly, a correspondence between the convergence of the target agreement rate and the target accuracy is seen, which validates our hypothesis that $A$ can be used

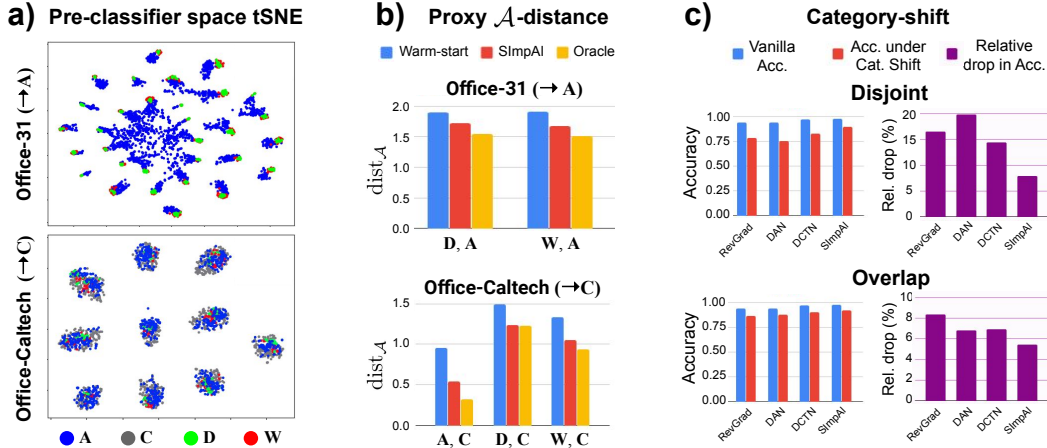

Figure 4: **(a) t-SNE.** We show alignment of domains at the pre-classifier space ($f$-output). **(b) Proxy $\mathcal{A}$-distance ($\downarrow$).** The values of $\mathrm{dist}_{\mathcal{A}}$ (Sec. 4.2a) are obtained between each source-target pair for the corresponding models shown in (a). **(c) Performance drop under category-shift ($\downarrow$).** Following [61], we compare SImpAl against RevGrad [10], DAN [29], DCTN [61] in the Overlap and Disjoint scenarios on Office-31 (**A, D → W**).

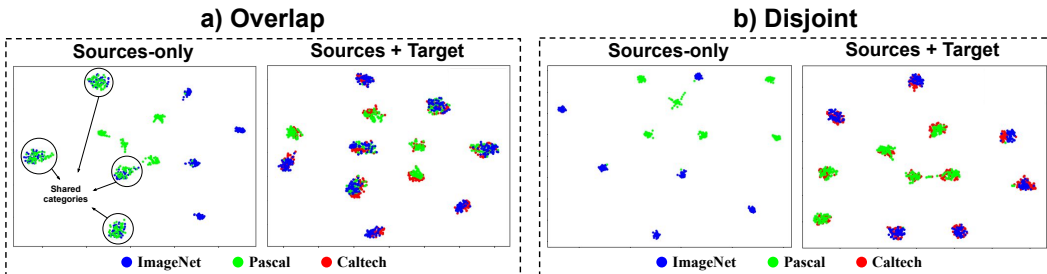

Figure 5: Pre-classifier space t-SNE embeddings for category-shift scenarios on ImageCLEF (**I, P → C**). Two plots are shown: features of the source domains only, and, features of all the domains. Best viewed in color. **(a) Overlap.** Here, 4 classes are shared between the sources ($|\mathcal{C}_{s_1} \cap \mathcal{C}_{s_2}| = 4$). Observe that the shared classes align, while the domain-specific classes are clustered separately. **(b) Disjoint.** In the absence of shared classes ($\mathcal{C}_{s_1} \cap \mathcal{C}_{s_2} = \phi$), class-wise alignment is not observed among the sources, which is essential to avoid negative-transfer. Note, in both cases, the target clusters align with the respective source clusters (since $\mathcal{C} = \mathcal{C}_{s_1} \cup \mathcal{C}_{s_2}$). This demonstrates implicit alignment under category-shift. See Suppl. for wider trends.

as a cue to determine the training convergence. This result is of interest in Unsupervised Domain Adaptation methods where the requirement of target labels has been the de-facto for model selection.

**d) Do the classifiers agree on correct pseudo-labels?** We also calculate the classifier agreement (and disagreement) for target samples that are pseudo-labeled correctly. Notably, Fig. 7a demonstrates that the classifiers tend to agree on an increasing number of target samples with correct pseudo-label predictions. This motivates the periodic update of $\mathcal{D}_t{}'$ (Lines 12-13 in Algo. 1), which captures an increasing number of target samples with correct pseudo-labels, as the adaptation proceeds.

**e) How accurate are target pseudo-labels?** As described in Sec. 3.2, we use classifier agreement to select target samples ($\mathcal{D}_t{}'$) with a higher pseudo-label accuracy. In Fig. 6b, we plot the accuracy of pseudo-labels separately for target samples having classifier agreement (*i.e.* $a(\mathbf{x}_t, f, h) = 1$) and disagreement (*i.e.* $a(\mathbf{x}_t, f, h) = 0$). Clearly, pseudo-labels are more accurate (more reliable) when the classifiers agree. Further, the accuracy on the target samples with agreement, $\mathcal{D}_t{}'$, is higher than the accuracy on all target samples, $\mathcal{D}_t$ (orange curve in Fig. 6b). Thus, the use of $\mathcal{D}_t{}'$ with a higher accuracy in pseudo-labels plays a key role in gradually improving the target performance.

**f) Using curriculum for target samples.** We form a curriculum for the target samples using the average classifier margin $w(\mathbf{x}_t, f, h)$ as a weight. Fig. 7c shows the geometrical interpretation of $w$, that measures how far into the agreement region a target sample falls. Thus, $w$ can be seen as a measure of the confidence in the prediction. As studied by prior methods [15, 45, 53], high confidence predictions are often correct. We show this in Fig. 7b where we plot the precision of

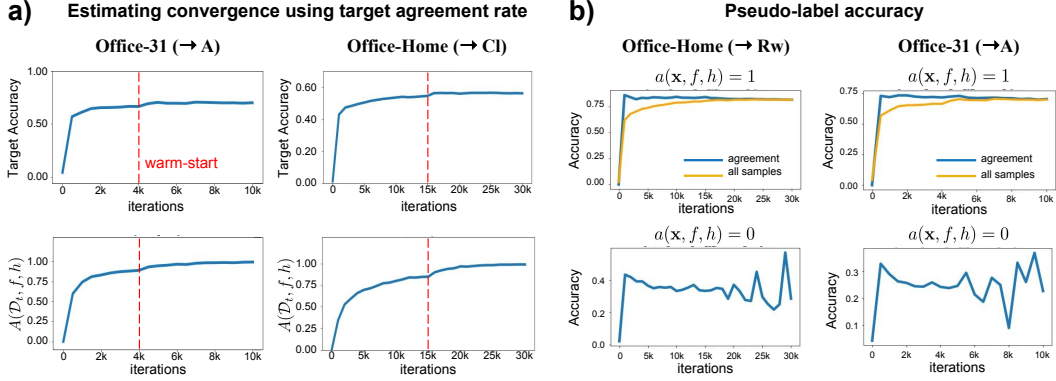

Figure 6: **(a) Target agreement rate** $A(\mathcal{D}_t, f, h)$**.** The target agreement rate (bottom) can be used as a cue to determine training convergence (top). **(b) Pseudo-label accuracy.** We observe a higher pseudo-label accuracy for target samples with $a(\mathbf{x}_t, f, h) = 1$ (top, blue curve), as compared to those with $a(\mathbf{x}, f, h) = 0$ (bottom).

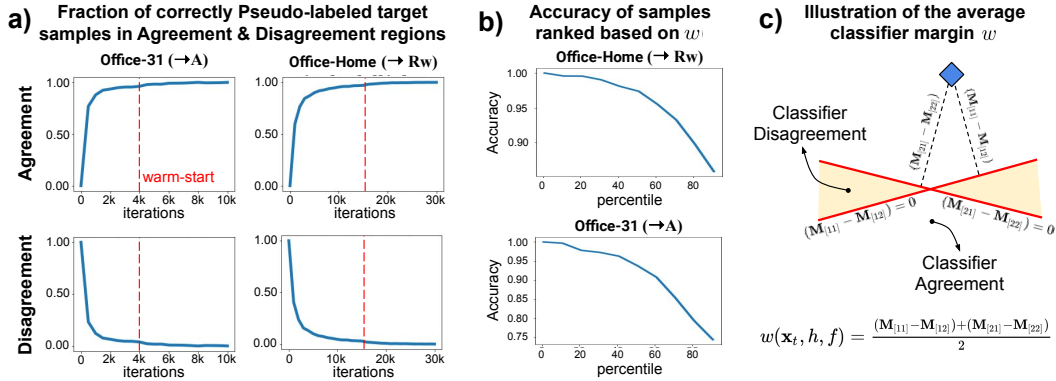

Figure 7: **(a) Migration of target samples with correct pseudo-labels.** During training, we find that the fraction of target samples with correct pseudo-label predictions increases in the agreement region. **(b) Curriculum using** $w(\mathbf{x}_t, f, h)$**.** When sorted in descending order based on $w$, the target samples exhibit an easy-to-hard curriculum. **(c) Geometrical interpretation of** $w$**.** The figure shows the significance of $w$ in a two-source two-class scenario (as in Fig. 1). Intuitively, $w$ is a measure of classifier confidence averaged over each classifier. Target samples that are further into the classifier agreement region exhibit a higher value of $w$.

target pseudo-labels at various confidence percentiles (in descending order of $w$). The accuracy shows a decreasing trend with $w$, validating our hypothesis that $w$ yields an easy-to-hard curriculum. Although our framework supports confidence thresholding to further minimize the pseudo-label noise, we do not employ thresholds for the main results (Table 1) as it introduces sensitive hyperparameters. See Supplementary for an empirical analysis with confidence thresholding.

## 5 Conclusion

In this paper, we demonstrated Self-supervised Implicit Alignment (SImpAl), that serves as a simple method to perform Multi-Source Domain Adaptation (MSDA). We observed that deep models exhibit the potential to implicitly align features under label supervision, even in the presence of domain-shift. We demonstrated the use of classifier agreement in SImpAl - to obtain pseudo-labeled target samples, to perform latent space alignment and to determine the training convergence. Extensive empirical analysis demonstrates the efficacy of SImpAl for MSDA.

Our work can facilitate the study of simple and effective algorithms for unsupervised domain adaptation. The insights obtained from our study can be used to explain the efficacy of a number of related self-supervised approaches. A potential direction of research is to develop efficient adaptation algorithms that are devoid of sensitive hyperparameters. Exploring SImpAl for scenarios such as Universal Domain Adaptation [63] would also be of future interest.

## Broader Impact

This work presents a simple and effective solution for Multi-Source Domain Adaptation, that has a two-fold positive impact. First, the method is aimed at improving the performance of prediction models by mitigating the bias caused by domain-shift between the training dataset and the test data encountered when deployed in a real-world environment. This is of growing interest in the machine learning community. Secondly, the insights presented in this work facilitate the study of efficient methods to perform domain adaptation, motivating the innovation of, for instance, energy-efficient methods to generalize deep models. While the method shows promising results under domain-shift, one should be cautious of the use of the pseudo-labeling procedure in the presence of adversarial samples, where the pseudo-labels may be less reliable and may result in performance degradation.

## Acknowledgments and Disclosure of Funding

This work was supported by a project grant from MeitY (No.4(16)/2019-ITEA), Govt. of India and a WIRIN project. We would also like to thank the anonymous reviewers for their valuable suggestions.

## Footnotes

[1] http://val.cds.iisc.ac.in/simpal

[2] http://imageclef.org/2014/adaptation.

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
