[Supplementary Material]

# Supplementary: Your Classifier can Secretly Suffice Multi-Source Domain Adaptation

## 1 Results on DomainNet

Owing to the limits of space, we present a summary of results on *DomainNet* in the paper. The full comparison is presented here in Table 1. The results for the prior arts are reported from [9].

Table 1: Results on the *DomainNet* [9] dataset.

|  | Method | →Clp | →Inf | →Pnt | →Qdr | →Rel | →Skt | Avg |
|---|---|---|---|---|---|---|---|---|
| SB | Source Only | 39.6±0.5 | 8.2±0.7 | 33.9±0.6 | 11.8±0.6 | 41.6±0.8 | 23.1±0.7 | 26.4±0.7 |
|  | DAN [5] | 39.1±0.5 | 11.4±0.8 | 33.3±0.6 | 16.2±0.3 | 42.1±0.7 | 29.7±0.9 | 28.6±0.6 |
|  | RTN [7] | 35.3±0.7 | 10.7±0.6 | 31.7±0.8 | 13.1±0.6 | 40.6±0.5 | 26.5±0.7 | 26.3±0.7 |
|  | JAN [6] | 35.3±0.7 | 9.1±0.6 | 32.5±0.6 | 14.3±0.6 | 43.1±0.7 | 25.7±0.6 | 26.7±0.6 |
|  | ADDA [12] | 39.5±0.8 | 14.5±0.6 | 29.1±0.7 | 14.9±0.5 | 41.9±0.8 | 30.7±0.6 | 28.4±0.7 |
|  | MCD [11] | 42.6±0.3 | 19.6±0.7 | 42.6±0.9 | 3.8±0.6 | 50.5±0.4 | 33.8±0.8 | 32.2±0.6 |
| SC | Source Only | 47.6±0.5 | 13.0±0.4 | 38.1±0.4 | 13.3±0.3 | 51.9±0.8 | 33.7±0.5 | 32.9±0.5 |
|  | DAN [5] | 45.4±0.4 | 12.8±0.8 | 36.2±0.5 | 15.3±0.3 | 48.6±0.7 | 34.0±0.5 | 32.1±0.5 |
|  | RTN [7] | 44.2±0.5 | 12.6±0.7 | 35.3±0.5 | 14.6±0.7 | 48.4±0.6 | 31.7±0.7 | 31.1±0.6 |
|  | JAN [6] | 40.9±0.4 | 11.1±0.6 | 35.4±0.5 | 12.1±0.6 | 45.8±0.5 | 32.3±0.6 | 29.6±0.5 |
|  | ADDA [12] | 47.5±0.7 | 11.4±0.6 | 36.7±0.5 | 14.7±0.5 | 49.1±0.8 | 33.5±0.4 | 32.2±0.6 |
|  | MCD [11] | 54.3±0.6 | 22.1±0.7 | 45.7±0.6 | 7.6±0.4 | 58.4±0.6 | 43.5±0.5 | 38.5±0.6 |
| MS | DCTN [13] | 48.6±0.7 | 23.5±0.5 | 48.8±0.6 | 7.2±0.4 | 53.5±0.5 | 47.3±0.4 | 38.2±0.5 |
|  | M³SDA [9] | 57.2±0.9 | 24.2±1.2 | 51.6±0.4 | 5.2±0.4 | 61.6±0.8 | 49.6±0.5 | 41.5±0.7 |
|  | M³SDA-$\beta$ [9] | 58.6±0.5 | 26.0± 0.8 | 52.3±0.5 | 6.3±0.5 | 62.7±0.5 | 49.5±0.7 | 42.6±0.6 |
|  | $SImpAl_{101}$ | 66.4±0.8 | 26.5±0.5 | 56.6±0.7 | 18.9±0.8 | 68.0±0.5 | 55.5±0.3 | 48.6±0.6 |

## 2 Further Analysis on Implicit Alignment

In this section present further analysis on implicit alignment. First, we show wider trends for implicit alignment under category-shift. Then, we present ablations on the training objective (with and without classifier agreement, single classifier head). Finally, we study thresholding schemes.

### 2.1 Implicit Alignment under Category-shift

We find that SImpAl works well even under category-shift. In Fig. 4c of the paper, we compare the relative drop (%) in accuracy under category-shift which is obtained as,

$$\text{relative drop} = \frac{(A_{vanilla} - A_{category-shift})}{A_{vanilla}} \times 100 \qquad (1)$$

Figure 1: Pre-classifier space ($f$-output) t-SNE showing the alignment of features under the category-shift settings **Overlap** (top) and **Disjoint** (bottom) on the ***Office-Caltech*** dataset for the task **A, D, W→C**. Note the alignment of selected clusters (shared classes among the source domains) in the **Overlap** scenario, while none of the source clusters align in the **Disjoint** scenario. Best viewed in color.

Figure 2: Pre-classifier space t-SNE showing the alignment of features under the category-shift settings **Overlap** (top) and **Disjoint** (bottom) on the ***ImageCLEF*** dataset for the task **I, P→C**. Best viewed in color.

Our approach exhibits a relatively lower drop in accuracy. To understand this better, we further investigate the latent space alignment of the domains after adaptation using SImpAl. We show two t-SNE plots, 1) Sources-only, *i.e.* showing alignment among the source domains, and 2) Sources + Target, *i.e.* showing the alignment among all the domains, corresponding to an adapted model.

Fig. 1 shows the latent space alignment for the task **A, D, W→C** for the *Office-Caltech* dataset. For the **Overlap** scenario, we set the number of shared categories to 2, while the number of source-private categories are 3, 3, 2 for the sources Amazon (**A**), DSLR (**D**) and Webcam (**W**) respectively. Clearly, alignment among the sources is observed only for the two shared categories (annotated in Fig. 1). In contrast, for the **Disjoint** scenario, the source domains **A**, **D**, **W** contain 3, 3, 4 unique classes respectively. Here, none of the source clusters align as expected (since each source has a distinct set of classes). However, in both scenarios, we find that the target domain aligns with at least one source

Figure 3: Pre-classifier space t-SNE showing the alignment of features under the category-shift settings **Overlap** (top) and **Disjoint** (bottom) on the *Office-31* dataset for the task **A, D→W** (in line with Fig. 4c of the paper). Source clusters corresponding to the shared classes are encircled. Best viewed in color.

domain (multiple sources if categories are shared). We find a similar trend across other datasets as shown in Fig. 2 for the **I, P→C** task of *ImageCLEF* (**Overlap**: 4 shared and 4, 4 source-private classes; **Disjoint**: 6, 6 source-private classes), and in Fig. 3 for the **A, D→W** task of *Office-31* (**Overlap**: 11 shared and 10, 10 source-private classes; **Disjoint**: 16, 15 source-private classes).

These results support the observation that implicit alignment can be leveraged even under category-shift. Note that, in methods such as moment matching [9] or adversarial alignment [2], there is usually no enforcement of class-level alignment across the domains. Thus, such methods are prone to negative transfer via conditional mis-alignment [4, 8], *i.e.* alignment of dissimilar classes. However, SImpAl enables class-conditional alignment by virtue of the label supervision, which could explain the relatively lower drop in performance under category-shift.

## 2.2   Is classifier agreement necessary?

We now turn towards the key observation that we present in the paper. By enforcing classifier agreement we are able to perform adaptation, while, when we learn domain-specific classifiers following previous MSDA approaches [13, 14], the latent space alignment is not seen. Here, we provide further empirical analysis in support of the observation through ablations on SImpAl$_{50}$.

**a) Warm-start with domain-specific classifiers.** We modify the loss formulation in Eq. 3 of the paper, to train domain-specific classifiers, as follows:

$$\min_{f,h} \quad \mathbb{E}_{\mathcal{D}\in\{\mathcal{D}_{s_{i'}}\}_{i'=1}^{n_d}} \quad \mathbb{E}_{(\mathbf{x}_{s_i}^{c_j}, y_{s_i}^{c_j})\in\mathcal{D}} \quad -\log(\sigma(\mathbf{M}_{[i\cdot]})_{[j]}) \tag{2}$$

where $\sigma$ denotes the softmax activation function. Essentially, an instance $\mathbf{x}_{s_i}^{c_j}$ pertaining to the source domain $\mathcal{D}_{s_i}$ trains the corresponding classifier head (note that the logarithmic term contains the probability of class $c_j$ of the classifier corresponding to $s_i$). This is in line with the prior methods such as [3, 13] where the domain-specific classifiers progressively learn to discriminate among the classes in their respective domains. Thus, there is no explicit enforcement of classifier agreement.

We perform warm-start with the loss formulation in Eq. 2 (name this model, "w/o agreement") and compare against another model learned using Eq. 3 of the paper (name this model, "with agreement"). Both models are trained for the same number of iterations under identical conditions. Finally, we test each model's performance on the target domain at warm-start. The multi-run statistics over 3 random seeds are presented in Table 2. The model "with agreement" shows a consistent improvement in performance in each scenario, which suggests that the model generalizes better to the target domain.

Table 2: Target accuracy of warm-start models with ablation on the learning approach - "w/o agreement" (learning domain-specific classifiers) vs. "with agreement" (our approach). Refer to Sec. 2.2a for discussion.

| Model | Office-31 ($\rightarrow$A) | ImageCLEF ($\rightarrow$P) | Office-Caltech ($\rightarrow$C) | Office-Home ($\rightarrow$Ar) |
|---|---|---|---|---|
| w/o agreement | $65.8 \pm 0.3$ | $75.8 \pm 0.3$ | $89.2 \pm 0.3$ | $67.3 \pm 0.7$ |
| with agreement | $66.2 \pm 0.3$ | $77.0 \pm 0.6$ | $90.3 \pm 0.4$ | $68.5 \pm 0.5$ |

Table 3: Proxy $\mathcal{A}$-distance ($\downarrow$) measured between each pair of domains for the warm-start models "with agreement", "w/o agreement" and "single classifier". Note that the model "with agreement" consistently exhibits lower $\text{dist}_{\mathcal{A}}$ than "with agreement", suggesting that it aligns the domains to a greater extent (Sec. 2.2a). Furthermore, we perform an ablation by using a single classifier head which also shows lower $\text{dist}_{\mathcal{A}}$ (Sec. 2.2b).

| Model | Office-31 ($\rightarrow$A) | | | ImageCLEF ($\rightarrow$ P) | | |
|---|---|---|---|---|---|---|
| | $\text{dist}_{\mathcal{A}}$(**A, D**) | $\text{dist}_{\mathcal{A}}$(**A, W**) | $\text{dist}_{\mathcal{A}}$(**D, W**) | $\text{dist}_{\mathcal{A}}$(**P, C**) | $\text{dist}_{\mathcal{A}}$(**P, I**) | $\text{dist}_{\mathcal{A}}$(**I, C**) |
| w/o agreement | 1.96±0.01 | 1.96±0.01 | 1.70±0.03 | 1.25±0.03 | 0.38±0.02 | 0.94±0.02 |
| with agreement | 1.93±0.00 | 1.93±0.00 | 0.56±0.04 | 0.65±0.04 | 0.34±0.01 | 0.65±0.04 |
| single classifier | 1.91±0.03 | 1.91±0.02 | 0.52±0.2 | 1.04±0.12 | 0.38±0.06 | 0.63±0.11 |

| Model | Office-Home ($\rightarrow$ Ar) | | | | | |
|---|---|---|---|---|---|---|
| | $\text{dist}_{\mathcal{A}}$(**Ar, Rw**) | $\text{dist}_{\mathcal{A}}$(**Ar, Cl**) | $\text{dist}_{\mathcal{A}}$(**Ar, Pr**) | $\text{dist}_{\mathcal{A}}$(**Cl, Pr**) | $\text{dist}_{\mathcal{A}}$(**Cl, Rw**) | $\text{dist}_{\mathcal{A}}$(**Pr, Rw**) |
| w/o agreement | $0.70 \pm 0.02$ | $1.46 \pm 0.05$ | $1.24 \pm 0.04$ | $1.36 \pm 0.07$ | $1.38 \pm 0.06$ | $0.65 \pm 0.00$ |
| with agreement | $0.64 \pm 0.01$ | $1.04 \pm 0.07$ | $0.98 \pm 0.02$ | $0.69 \pm 0.04$ | $0.75 \pm 0.02$ | $0.36 \pm 0.03$ |
| single classifier | 0.61±0.02 | 1.16±0.10 | 1.06±0.06 | 0.90±0.15 | 0.96±0.14 | 0.44±0.06 |

| Model | Office-Caltech ($\rightarrow$ C) | | | | | |
|---|---|---|---|---|---|---|
| | $\text{dist}_{\mathcal{A}}$(**C, A**) | $\text{dist}_{\mathcal{A}}$(**C, D**) | $\text{dist}_{\mathcal{A}}$(**C, W**) | $\text{dist}_{\mathcal{A}}$(**A, D**) | $\text{dist}_{\mathcal{A}}$(**A, W**) | $\text{dist}_{\mathcal{A}}$(**D, W**) |
| w/o agreement | $1.05 \pm 0.01$ | $1.80 \pm 0.00$ | $1.78 \pm 0.02$ | $1.94 \pm 0.01$ | $1.92 \pm 0.02$ | $1.61 \pm 0.06$ |
| with agreement | $0.91 \pm 0.02$ | $1.49 \pm 0.05$ | $1.38 \pm 0.04$ | $1.67 \pm 0.01$ | $1.49 \pm 0.02$ | $0.92 \pm 0.05$ |
| single classifier | $0.97 \pm 0.05$ | $1.63 \pm 0.09$ | $1.55 \pm 0.15$ | $1.77 \pm 0.14$ | $1.66 \pm 0.19$ | $1.05 \pm 0.18$ |

To uncover the underlying effect, we measure the Proxy $\mathcal{A}$-distance [1] defined as $\text{dist}_{\mathcal{A}} = 2(1 - 2\epsilon)$ where $\epsilon$ is the generalization error of a domain discriminator. A lower value of this measure indicates a higher degree of alignment between the domains. In Table 3, we report $\text{dist}_{\mathcal{A}}$ between each pair of domains at the $f$-output space (multi-run statistics corresponding to the models trained in Table 2). Clearly, our approach exhibits a higher degree of alignment between the source domains, as compared to learning source-specific classifiers. This encourages the model to learn domain-agnostic features that are more generalizable to the target domain, resulting in an improved alignment between each source-target pair. This translates to an improvement in the target performance.

**b) Using a single classifier head.** We also train a model by replacing multiple classifiers having agreement with a single classifier. In this case, we fix the number of iterations to be the same as those obtained from SImpAl$_{50}$. In Table 4, we compare the adaptation results of the single-classifier model, against our approach (with agreement using $n_d$ classifiers). Clearly, the performance is better when using multiple classifier heads with agreement. For the single-classifier model, we find that during adaptation the performance reaches a peak and then marginally declines to a lower value before reaching the maximum number of iterations, perhaps due to noisy pseudo-labels [13]. In our approach however, classifier agreement aids in pruning those noisy samples near the decision boundaries and enhances the pseudo-labels. This is the added benefit of using multiple classifiers.

To study the extent of latent space alignment using a single classifier, we measure the Proxy $\mathcal{A}$-distance. In Table 3, we present the $\text{dist}_{\mathcal{A}}$ values at warm-start for the single classifier model and compare it against the two aforementioned approaches ("with" and "w/o" agreement). Particularly, we find that the $\text{dist}_{\mathcal{A}}$ values are either similar to the model "with agreement", or lower than the model "w/o agreement", indicating that even a single classifier enables alignment to an extent. This explains the efficacy of self-supervised approaches that use pseudo-labels for training under domain-shift.

## 2.3 Ablations using thresholding schemes

In self-training based approaches, pseudo-labeled samples are usually obtained using confidence-thresholding, *i.e.* those samples exhibiting a confidence above a certain threshold $\tau$ are chosen for self-training. However, this results in a sensitive hyperparameter $\tau$ which requires labeled target

Table 4: Target adaptation performance for models trained with a single classifier head ($n = 1$) and our approach with $n = n_d$ classifier heads. Refer to Sec. 2.2b for discussion.

| No. of Heads | Office-31 ($\rightarrow$A) | ImageCLEF ($\rightarrow$P) | Office-Caltech ($\rightarrow$C) | Office-Home ($\rightarrow$Ar) |
|---|---|---|---|---|
| $n = 1$ | $68.8 \pm 0.6$ | $75.9 \pm 0.7$ | $89.8 \pm 1.3$ | $68.4 \pm 0.7$ |
| $n = n_d$ | $70.6 \pm 0.6$ | $77.5 \pm 0.3$ | $92.2 \pm 0.1$ | $70.8 \pm 0.2$ |

Table 5: Target adaptation performance using softmax confidence based thresholding ($\tau$) and percentile based bagging ($\gamma$) schemes in our approach. Refer to Sec. 2.2b for discussion.

| Threshold | Office-31 ($\rightarrow$A) | ImageCLEF ($\rightarrow$P) | Office-Caltech ($\rightarrow$C) | Office-Home ($\rightarrow$Ar) |
|---|---|---|---|---|
| $\tau = 0.80$ | $70.4 \pm 0.8$ | $76.6 \pm 0.4$ | $92.4 \pm 1.1$ | $71.5 \pm 0.5$ |
| $\tau = 0.75$ | $70.8 \pm 0.2$ | $76.8 \pm 1.0$ | $93.3 \pm 0.1$ | $72.2 \pm 0.3$ |
| $\tau = 0.70$ | $71.8 \pm 1.3$ | $77.7 \pm 0.4$ | $92.8 \pm 0.7$ | $71.9 \pm 0.4$ |
| $\tau = 0.65$ | $72.6 \pm 0.5$ | $77.1 \pm 0.7$ | $92.4 \pm 1.3$ | $72.2 \pm 0.5$ |
| $\tau = 0.60$ | $71.4 \pm 1.0$ | $77.7 \pm 0.8$ | $92.5 \pm 1.3$ | $72.4 \pm 0.5$ |
| $\tau = 0.55$ | $73.4 \pm 0.7$ | $78.1 \pm 0.5$ | $93.1 \pm 1.0$ | $71.9 \pm 0.2$ |
| $\tau = 0.50$ | $72.5 \pm 1.0$ | $77.4 \pm 0.9$ | $92.3 \pm 0.4$ | $71.2 \pm 0.1$ |
| $\gamma = 5\%$ | $69.3 \pm 0.2$ | $77.4 \pm 0.9$ | $92.7 \pm 0.3$ | $73.2 \pm 0.4$ |
| $\gamma = 10\%$ | $70.3 \pm 0.9$ | $77.9 \pm 0.2$ | $92.8 \pm 0.7$ | $72.6 \pm 0.3$ |
| $\gamma = 15\%$ | $71.9 \pm 0.8$ | $78.6 \pm 0.7$ | $92.1 \pm 0.3$ | $72.8 \pm 0.6$ |
| SImpAl$_{50}$ | $70.6 \pm 0.6$ | $77.5 \pm 0.3$ | $92.2 \pm 0.1$ | $70.8 \pm 0.2$ |

samples for tuning appropriately. In our framework, we propose $w(\mathbf{x}_t, f, h)$, defined in Eq. 6 of the paper, as a measure of confidence in pseudo-label prediction for target instances, that results in an easy-to-hard curriculum (Sec. 4.2f in the paper). While our framework supports confidence based thresholding, we do not use it for the main results in the paper since it introduces additional hyperparameters. Here, we present empirical results by incorporating thresholds in SImpAl$_{50}$.

We incorporate two types of thresholds, softmax-confidence based (similar to Saito *et al.* [10]) and percentile-based bagging (top confident samples based on $w$). In both cases, target samples are first filtered based on classifier agreement, *i.e.* $\mathcal{D}_t' = \{(\mathbf{x}_t, y_t^{c_j}) \mid \mathbf{x}_t \in \mathcal{D}_t,\ a(\mathbf{x}_t, f, h) = 1\}$, and the samples are further chosen based on thresholding or bagging schemes.

For the softmax-confidence based threshold ($\tau$), we follow the method applied in Saito *et al.* [10], where a prediction is considered confident if at least one of the classifiers exhibit the $\arg\max$-confidence above a threshold $\tau$. Intuitively, self-training with such confident target samples will encourage more target samples to fall in the high confidence region. For the percentile-based bagging scheme, we choose the top $\gamma$-percentile (most confident) target samples from $\mathcal{D}_t'$ for pseudo-labeling. At each pseudo-label update, the percentile is increased by $\gamma$, *i.e.* the pseudo-labeled bag progressively grows in integral steps of $\gamma$. Both methods are trained for the same number of iterations as SImpAl$_{50}$.

Table 5 shows that self-training is sensitive to the confidence threshold $\tau$ and bagging percentile $\gamma$. Although the adaptation performance is seen to improve using thresholds, the best hyperparameter values are highly dataset specific. This calls for labeled target samples to reliably establish the most appropriate hyperparameter values for the task at hand. The study of automated methods to select threshold hyperparameters using unlabeled target samples would be of interest.

## 3 Practical Application: Cross-Domain Image Retrieval

We now demonstrate a practical application of Implicit Alignment. Consider an unlabeled dataset $\mathcal{D}$ of images sampled from a target domain. Suppose we wish to retrieve images from this dataset based on some class semantics (for *e.g.*, "retrieve images of objects having wheels"). With an unannotated dataset, this task seems non-trivial. However, we show that this is possible by performing Multi-Source Domain Adaptation on the target dataset $\mathcal{D}$ using SImpAl. Considering $\mathcal{D}$ as the target domain, we adapt a deep model using labeled source datasets and the unlabeled target data $\mathcal{D}$, under the SImpAl framework, and use this model to measure semantic similarity.

---
**Algorithm 1** Cross-Domain Image Retrieval
---
1: **require:** Reference image $\mathbf{x}_r$, Query set $\mathcal{D}$, Model $\{f, h\}$        $\triangleright \cup$ denotes union

    $\triangleright$ Calculate the class probability vector for $\mathbf{x}_r$ (using Eq. 1 of the paper)
2:   $\mathbf{M} \leftarrow h \circ f(\mathbf{x}_r)$
3:   $\mathbf{p}_r \leftarrow \frac{1}{n_d} \sum_{i=1}^{n_d} \sigma(\mathbf{M}_{[i\cdot]})$

    $\triangleright$ Calculate class probability vectors for images in the query set $\mathcal{D}$ (using Eq. 1 of the paper)
4:   $\mathcal{R} \leftarrow \{ \}$ $\triangleright$ Create an empty collection of distance values
5:   **for** $\mathbf{x}_q \in \mathcal{D}$ **do**
6:       $\mathbf{M} \leftarrow h \circ f(\mathbf{x}_q)$
7:       $\mathbf{p}_q \leftarrow \frac{1}{n_d} \sum_{i=1}^{n_d} \sigma(\mathbf{M}_{[i\cdot]})$
8:       $\mathcal{R} \leftarrow \mathcal{R} \cup \{\|\mathbf{p}_q - \mathbf{p}_r\|_2\}$ $\triangleright$ get the euclidean distance in class probabilities
9:   $\mathcal{D}' \leftarrow$ set of $k$ nearest images in the query set $\mathcal{D}$ based on the distance $\mathcal{R}$

10: **return** $\mathcal{D}'$
---

Figure 4: Sample images in the Quickdraw (**Qdr**) and Real (**Rel**) domains of the *DomainNet* [9] dataset. Note, this figure is meant for qualitatively demonstrating the domain-shift; the images do not correspond.

We develop a cross-domain image retrieval system where, given a reference image $\mathbf{x}_r$, we retrieve semantically similar images from a given set of images $\mathcal{D}$ (called the query set). To achieve this, we obtain the nearest neighbors of the reference image $\mathbf{x}_r$ in the set $\mathcal{D}$, based on the class probability vector $\mathbf{p}$ (as obtained in Eq. 1 of the paper). This is shown in Algorithm 1.

Note, Algo. 1 does not require any label information. Thus, the query set $\mathcal{D}$ can be unlabeled. We show an example use-case where we consider images in the "Quickdraw" (**Qdr**) domain of *DomainNet* [9] as the reference images, while the query set is the unlabeled target domain "Real" (**Rel**). Fig. 4 shows sample images in the two domains, which exhibit a large domain-shift.

We obtain a model adapted to the "Real" target domain (*i.e.* the model corresponding to →**Rel** in Table 1). Given a reference image in the **Qdr** domain, an end-user can retrieve semantically similar images from the unlabeled **Rel** domain using Algo. 1 above.

In Fig. 5, we present qualitative results, demonstrating the retrieval of images from the **Rel** domain using randomly selected images from the **Qdr** domain as reference. Note that, all images in Fig. 5 pertain to the test set of the corresponding domains that the model has not encountered during adaptation. The retrieval process using Algo. 1 above works surprisingly well, yielding qualitatively satisfactory results (annotated with green tick-mark). While there are certain images which yield false retrievals (annotated with red cross-mark), many of those cases have incomprehensible reference image (marked with an orange question-mark) and can be ignored during qualitative evaluation.

Further, we design a tool to retrieve images from a chosen query set, by manually "doodling" class semantics (similar to the images in the **Qdr** domain). See code implementation for the demonstration. Fig. 6 shows the images retrieved during the demonstration.

# 4 Code Reference

Pytorch implementation (with cross-domain image-retrieval demo) can be found on the project page[1].

Figure 5: Images retrieved from the Real (**Rel**) domain (below) using reference images from the Quickdraw (**Qdr**) domain (above). The model corresponds to the →**Rel** task of *DomainNet*. Here, we show the nearest neighbor ($k = 1$ in Line 9 of Algo. 1 above). All images pertain to the test set of *DomainNet*. Green tick-marks indicate satisfactory retrievals, red cross-marks indicate false retrievals and orange question-marks indicate the cases where the reference image is incomprehensible. Note the high success rate of retrievals.

Figure 6: An example showing the top-16 retrievals ($k = 16$ in Line 9 of Algo. 1 above) from the **Rel** domain by manually "doodling" objects (see demonstration video on the project page).

## Footnotes

[1] http://val.cds.iisc.ac.in/simpal