[Reviews · NeurIPS 2020]

Review 1

Summary and Contributions: This paper proposes a simple yet efficient self-training scheme for multi-source domain adaptation. It enforces an agreement among the classifiers to align the domains in the latent space. Specifically, it uses pseudo-labeled target samples and enforces the classifiers to agree upon the pseudo-labels The extensive evaluations are provided in the experiment part.

Strengths: This paper proposes a simple yet efficient self-training scheme for multi-source domain adaptation. Clear writing for the method part. It is easy to understand. It provides a thorough evaluation of our approach on five benchmarks with detailed ablation study.

Weaknesses: My major concern is the novelty of the self-training based domain adaptation. The self-training DA is well established, and the adaptation from two domains (source/target) is somewhat off-the-shelf to the multi-source case. [a]Confidence regularized self-training. ICCV 19 [b]Unsupervised domain adaptation for semantic segmentation via class-balanced self-training. ECCV 18 Considering the previous work, the contribution of this paper seems limited. It is not clear why the additional two-stage training is better than the alternation of source/target training in each iteration in [a,b]. Moreover, the authors do not incoperate these two important prior work for discussion. A more detailed justification of "samples with a higher w are more likely to possess correct pseudo-labels." can be helpful. It would be interesting to investigate how to control the pretty noisy pseudo-label in this framework.

Correctness: It is technically sound.

Clarity: I am satisfied with the clarity of the method parts.

Relation to Prior Work: In sufficient. Especially for the self-training based domain adaptation. [a]Confidence regularized self-training. ICCV 19 [b]Unsupervised domain adaptation for semantic segmentation via class-balanced self-training. ECCV 18

Reproducibility: Yes

Additional Feedback:


Review 2

Summary and Contributions: This paper addresses the problem of multi-source domain adaptation. The goal of this task is to learn a model that transfers the knowledge learned from multiple labeled source domains to an unlabeled target domain. The authors formulate the adaptation as leveraging the feature alignment implicitly exhibited by the deep neural networks, and propose to leverage the pseudo labels of the target samples to learn the classifier in a self-supervised fashion. Experiments are conducted on five benchmarks, including the Office-Caltech, ImageCLEF, Office-31, Office-Home, and DomainNet datasets. Experimental results show some improvement over competing methods in some cases.

Strengths: 1. The idea of leveraging the implicit alignment exhibited by deep neural networks to address multi-source domain adaptation is interesting and makes sense. 2. Experiments are carried out extensively on five benchmarks. 3. The discussions in Section 4.2 are extensive and in detailed. The authors answer several important questions that a reader may have. In my opinion, it is rare to see a paper that has this kind of discussions. This is what I think the most valuable part for this submission (while many papers do not have this kind of discussions).

Weaknesses: 1. This paper lacks some very important references for domain adaptation. The authors should cite and discuss in the revised manuscript. - Li et al. Bidirectional Learning for Domain Adaptation of Semantic Segmentation. In CVPR, 2019. https://arxiv.org/pdf/1904.10620.pdf - Chen et al. CrDoCo: Pixel-level Domain Transfer with Cross-Domain Consistency. In CVPR, 2019. https://arxiv.org/pdf/2001.03182.pdf 2. My minor concern is that in Table 1 A and B, the proposed method seems to have inferior performance compared to MFSAN. It is a bit pity that the proposed method is not the state-of-the-art on these two datasets.

Correctness: Yes.

Clarity: Yes.

Relation to Prior Work: Yes.

Reproducibility: Yes

Additional Feedback: I really like the discussions in Section 4.2. The authors really "teach" me something. However, due to the inferior performance compared to MFSAN and the authors miss some important references for domain adaptation, I can only rate this paper 6 (marginally above the acceptance threshold). I will take a look at the rebuttal as well as the review comments from other reviewers and see if there is any other factor that makes me to change my mind. -- post rebuttal comments -- I have read all the review comments and the rebuttal. I agree with the points raised by Reviewer 3. However, I believe this paper has some value. Thus, I tend to remain my rating.


Review 3

Summary and Contributions: This paper considers the multi-source domain adaptation (MDA) problem and proposed a self-training method to solve it. The problem is meaningful and the improvement given by self-training is obvious.

Strengths: The paper is good written and the result is good

Weaknesses: 1. Figure 2 is very confusing to me. Figure (2a) seems to train an individual classifier for each domain. Figure (2b) seems also to train an individual classifier for each domain but also require the agreement across all classifiers for all samples. My question is what is the difference between Figure 2b and the method only train one classifier for all domains? It seems to me that training several classifiers and alignment them with a loss is same as only training one classifier. Furthermore, I agree that alignment classifiers can align features across domains as well, but the previous methods also use some distance loss or adversarial learning to align classifiers or features, which will reach a similar performance. But there is not any clear analysis about my concern. 2. The method used in this paper is quite trivial. After reading the paper, I found the author just proposed to use self-training technique to solve the MDA problem. I hardly think this technique is new. It has been used in a lot of domain adaptation papers such as [1][2]. In fact, I think self-training almost becomes a standard technique to improve the performance for domain-adaptation. Maybe the only difference given by this paper is they have multiple classifiers and they force the classifiers to generate same pseudo-labels. But this will come back to my first question. Why training multiple classifiers and align them is better than just training one classifier. 3. Although the author did a lot of experiment to analyze how the proposed self-training works, several key experiments are missing in my opinion. As this is a self-training paper, the author should compare their method with other self-training papers. The simplest one is [1] by using a threshold to pick up labels for target samples. Furthermore, I also want to know if the self-training can work with adversarial learning? In this paper, the author just remove all the discriminators without any explanation. [1] Domain Adaptation for Semantic Segmentation via Class-Balanced Self-Training [2] bidirectional learning for domain adaptation of semantic segmentation

Correctness: Yes

Clarity: Yes

Relation to Prior Work: Yes

Reproducibility: Yes

Additional Feedback: Overall speaking, the paper is less novel to me and the experiments only focusing on explaining the proposed method and lacks comparison with others. After reading the rebuttal and discussing with other reviewer, I still think the paper has very limited novelty. Thus I stay the same score.


Review 4

Summary and Contributions: This paper addresses the problem of multi-source domain adaptation. Most of existing methods rely on distribution matching schemes to learn domain invariant feature representations. This paper proposes a method from a different perspective. They use pseudo-labeled target samples and enforce the classifiers to agree upon the pseudo-labels, leveraging implicit alignment exhibited by deep models. The results by the proposed method are promising, showing that effectiveness of this training algorithm.

Strengths: This paper solves the multi-source domain adaptation problem from a different perspective. Discarding the idea of existing methods which need an auxiliary feature alignment loss to match distribution, this work proposes an effective and simple method. Under the observation that deep models implicitly aligning the latent features under supervision, they provide an interesting motivation to introduce target samples into the training process. The pseudo-labels are obtained from imposing an agreement of classifier predictions. Further, the authors conduct an extensive empirical evaluation of their approach over five benchmark datasets proving generalization to the target domain. The analysis of section 4.2 is comprehensive and credible.

Weaknesses: This work is similar to the DCTN[45] that the feature extractor and the category classifier are discriminatively fine-tuned with multi-source labeled and target pseudo-labeled data. However, this paper lacks the comparison analysis with DCTN. Since feature alignment and pseudo-label finetuned are both revising decision boundaries to adapt target domain. DCTN uses both of them. This paper should highlight the advantage as compared with DCTN.

Correctness: Both claims and methods are correct. The experimental results and ablation study are sufficient to prove the empirical methodology correct.

Clarity: The paper is well written and straightforward to understand.

Relation to Prior Work: To some extent, it is clear. However, as mentioned in Weakness, the comparation with DCTN is needed. Further, some more recent papers on multi-domain adaptation are missing, please refer to "Multi-source Domain Adaptation in the Deep Learning Era: A Systematic Survey" for details.

Reproducibility: Yes

Additional Feedback: As mentioned in Weakness, the pseudo-label part is consistent with DCTN, can you explain the superiority of your work? Since feature alignment and pseudo-label finetuned are both revising decision boundaries to adapt target domain. Why not combined them? Besides, As shown in Table1, all experimental setting has a pretty performance in source only, which means the domain gap between them is small. I am considering that when this method applying in the large domain gap, say the warm up stages is not worked for target. As a result, there are few pseudo-labeled target can be found. Will this method work? After rebuttal: I have read the other reviews and the rebuttal. I think the authors have addressed my concerns. Therefore, I'd like to keep my positive rating. The authors are suggested to revise the paper based on all the reviews.

[Author Response · NeurIPS 2020]

1 We thank the reviewers for their valuable suggestions to improve the draft. We address the concerns below.

**To all the reviewers.**

**a)** *"Novelty"*. Thanks for providing the citations to self-supervised approaches. Although most of them focus on solving specific challenges (that are different from MSDA), they do deserve a discussion, and we will include one in the revision. Our prime contribution is in the form of insights that lead to a simple design, which makes our work different. For *e.g.*, we find that classifier agreement ($A$) can hint at the migration of target samples during adaptation (Sec. 4.2c, 4.2e). Likewise, we show that even under category-shift (Sec. 4.2b), the model implicitly aligns only the shared classes across domains (Suppl. Sec. 2), which is relatively less explored in MSDA. Outperforming the state-of-the-art is not the objective here, rather, we wish to broaden the perspective of what deep models can achieve under domain-shift, to promote future research in this area. We will emphasize more on this aspect in the revision.

**Reviewer 1:**

**b)** *"Two stage training"*. Note, after the warm-start, we do have alternating source/target batches (see L8, L10 in Algo. 1, and implementation in Suppl.). The warm-start helps achieve reliable pseudo-labels before introducing target instances.

**c)** *"Justification of $w$"*. We empirically verify in Sec. 4.2f that $w$ roughly correlates with the accuracy of pseudo-labels.

**d)** *"Controlling the noisy pseudo-label"*. While the warm-start stage (Sec. 3.1), and the strategy in L139-L150 are intended to subdue the noise in pseudo-labels, one could also consider a subset of $\mathcal{D}_t{}'$ having the most confident target samples to further reduce the noise (see Fig. 6b) since correct labels are often predicted with high confidence [33,12,38].

**Reviewer 2:**

**e)** *"Comparison with MFSAN"*. MFSAN employs auxiliary loss functions at the feature level and at the output level for adaptation. In contrast, we harness implicit alignment exhibited by deep networks. While MFSAN is applicable in the closed-set scenario, SImpAl can be readily applied under category-shift (Sec. 4.2b, Suppl. Sec. 2). Further, MFSAN selects model based on best target performance, while ours is based on the convergence of agreement rate ($A$).

**Reviewer 3:**

**f)** *"Single Classifier"*. The reviewer's observation is correct. One can train a single classifier to exploit implicit alignment. However, there would be no way to measure classifier agreement ($a$) which has multiple benefits. Particularly, we find that classifier agreement indicates more accurate pseudo-labels (Fig. 5b), while also providing a convenient way to monitor the adaptation process (Fig. 5a). Since, multiple classifier heads enable the measurement of classifier agreement as a by-product (at no additional effort), we choose this approach. Note, while there is no upper limit to the number of classifier heads, employing too many heads delays the convergence of the agreement rate $A$ (due to a greater probability of disagreement), thereby requiring more iterations. Hence we fix the number of heads to be $n_d$.

**g)** *"Threshold based self-training / DCTN"*. Threshold based self-labeling has been carried out in DCTN [45]. However, as demonstrated in DCTN (see Fig. R1 below for reference), employing a threshold based pseudo-labeling alone results in performance degradation (blue curve) in such a method, where domain-specific classifiers are learned. Thus, an auxiliary adversarial alignment loss *is a requirement* in DCTN, to align the source and the target domains for adaptation. In contrast, SImpAl is motivated from the strong inductive bias of deep models to implicitly align the latent features under supervision (Fig. 2). The approach is based on enforcing classifier consistency (in contrast to learning domain-specific classifiers), which allows adaptation without requiring an explicit alignment loss.

**Reviewer 4:**

**h)** *"DCTN comparison"*. Please refer to point **(g)** above, and the discussion on category-shift in Sec. 4.2b and Suppl. L22-L26.

**i)** *"Larger domain-gap"*. The method also works under large domain gaps, as evaluated on the DomainNet dataset (see Table 1E, and Suppl. Table 1). Further, we demonstrate the alignment of latent features across domains which are vastly different (Quickdraw and Real-world images; see images in Suppl. Fig. 4). Specifically, we are able to perform cross-domain image retrieval (Suppl. Sec. 3). This tool is bundled with the Suppl. code, along with a demonstration video (image_retrieval_demo.mp4).

**Figure R1:** We show Fig. 4a of [45] (DCTN) here for reference (target accuracy vs. training progression). The blue curve corresponds to employing a pseudo-label only algorithm in DCTN (that learns domain-specific classifiers). The performance is observed to deteriorate. See Sec. 5.4 in DCTN [45] for discussion.

[Meta-Review · NeurIPS 2020]

This paper proposes a simple self-training mechanism for multi-source domain adaptation. In a pre-training step, the features extracted by the deep learning networks from different sources are aligned using pseudo labels generated by independent softmax classifiers. All the reviewers agree that one of the strong contributions of this paper is the detailed experimental analysis that provide good insights to the reader on the topic of multi-source domain adaptation. This compensates for the limited novelty in the components used in this model such as multiple softmax classifiers which have been employed in related context even if they are not specifically for multi-source domain adaptation. Reviewers have highlighted a few shortcomings that could be improved in the final version of the paper and the authors have agreed to doing so in the response.